# Simultaneous Connections Routing in Wavelength–Space–Wavelength Elastic Optical Switches

**DOI:** 10.3390/s23073615

**Published:** 2023-03-30

**Authors:** Enass Abuelela, Mariusz Żal, Wojciech Kabaciński

**Affiliations:** Institute of Communication and Computer Networks, Faculty of Computing and Telecommunication, Poznan University of Technology, 60-965 Poznan, Poland

**Keywords:** three-stage switching network, rearrangeable switching network, connection routing, elastic optical network

## Abstract

In this paper, we investigate the three-stage, wavelength–space–wavelength switching fabric architecture for nodes in elastic optical networks. In general, this switching fabric has r input and output switches with wavelength-converting capabilities and one center-stage space switch that does not change the spectrum used by a connection. This architecture is most commonly denoted by the WSW1 (*r*, *n*, *k*) switching network. We focus on this switching fabric serving simultaneous connection routing. Such routing takes place mostly in synchronous packet networks, where packets for switching arrive at the inputs of a switching network at the same time. Until now, only switching fabrics with up to three inputs and outputs have been extensively investigated. Routing in switching fabrics of greater capacity is estimated based on routing in switches with two or three inputs and outputs. We now improve the results for the switching fabrics with four inputs and outputs and use these results to estimate routing in the switching fabric with an arbitrary number of inputs and outputs. We propose six routing algorithms based on matrix decomposition for simultaneous connection routing. For the proposed routing algorithms, we derive criteria under which they always succeed. The proposed routing algorithms allow the construction of nonblocking switching fabrics with a lower number of wavelength converters and the reduction of the overall switching fabric cost.

## 1. Introduction

We can observe that in recent years elastic optical networks (EONs) are becoming more popular as a potential alternative to the rapidly growing popularity of optical networks. An example of this is the significant shift of ICT services to the new form, instead of dedicated fixed services. Software as a service (SaaS), function as a service (FaaS), infrastructure as a service (IaaS), and platform as a service (PaaS) are demonstrations of this new form [1,2,3]. Common features of these services include availability on demand, great scalability, adaptability, and flexibility [4,5]. This approach is reflected in the progress of data-transmission technologies and networks. For decades, optical domain technologies have been widely utilized in access and core networks, including wavelength division multiplexing (WDM), coarse WDM (CWDM), and dense WDM (DWDM). These technologies have significant limits in terms of scalability and flexibility. The International Telecommunication Union (ITU) has overcome the problem of defined fixed frequency grids by changing to a format that allows small sections of the optical spectrum to be selected and operated [6,7]. The concept of frequency slot units (FSUs) as an updated form of spectrum granularity was provided by this new bandwidth-allocation model. Aside from having a very small granularity (specified by [8] FSU is set to 12.5 GHz, 6.25 GHz, or smaller in width), this model has another feature. Any number of FSUs can be used in an optical connection, as long as the sum of the FSUs remains below the full spectrum and the FSUs are adjacent. In EONs, such a connection is a paradigm (known as the *m*-slot connection) [4,5,9,10]. A new switching fabric architecture is required for optical switching networks to create many different flexible paths that support a novel bandwidth-allocation model [4,5,9,11,12,13,14,15].

Basic information about the EON operation, including switch architectures and switching fabric functions, can be found in [16,17]. The problem of routing and spectrum allocation in EONs at the network level is surveyed in [18], while the spectrum fragmentation problems and management approaches were treated in [19]. In [20], the authors evaluated the performance of EONs when spectrum conversion is introduced in intermediate switches. The nonblocking two-stage switching fabrics with multirate connections and conversion in both stages were considered in [21,22]. The authors of [23] present the simulation studies of elastic optical switching fabrics based on the three-stage Clos switching fabric. They evaluated the loss probability for various traffic classes offered in a single optical network node. The multicast connections and wide-sense nonblocking conditions in optical WDM networks are considered in [24]. Finally, elastic optical switches are also considered for use in data center networks, where various architectures and nonblocking conditions were considered in [25,26,27].

In many papers that discuss the nonblocking properties of switching fabrics, space–wavelength–space (SWS) and wavelength–space–wavelength (WSW) are the two identified architectures. There are three stages in each architecture. One switch (SWS1/WSW1) or more than one switch (SWS2/WSW2) in the intermediate stage are the two categories of each architecture when classified in terms of the number of center-stage switches [12,13,14,15,25,28,29,30,31,32]. In the end, setting up a connection path by using the routing algorithm is one of the most important aspects. Two patterns are used to examine four node architectures employing waveband converters for EONs, according to allocation and distribution in [11]. In  [9], a switching node model is introduced that was found to meet the requirements of EONs as WSW switching fabrics. Similarly, the SWS1 and SWS2 architectures for flexible optical switching networks are described in [12]. The two WSW architectures (WSW1, WSW2) in [13], are investigated due to the lower number of FSUs in the interstage links, which can be provided by using more than one switch in the center stage (WSW2s). In [15], the authors considered the WSW1 switching fabric with 2, 3 and more input/output links and proposed seven routing algorithms.

In this paper, we define and illustrate the operation of six routing algorithms and derive sufficient conditions under which the proposed algorithms always end with success. The analysis shows that only three of the six algorithms are necessary for further study, and the required number of frequency slots in the interstage link can be smaller than that required in [15]. The construction cost, which is determined by the wavelength converters number, reflects this. As a result, the cost of building the switching nodes by using our proposed algorithm is less than the cost of constructing the switching nodes by using some of the previously published algorithms. Consequently, our algorithm succeeds if, for r=4, each interstage link contains only n+⌊2n3⌋ FSUs, while it requires 2n FSUs, in [15]. In the case of switching fabrics with r=4, it means that the proposed work is new and improved. A demonstration of the general formula and a detailed analysis of the proposed algorithm are included in the subsequent sections of the paper.

## 2. Rearrangeable Switching Fabric Architecture and Operation

### 2.1. Switching Fabric Architecture

In this work, we consider the WSW1(n,r,k) switching fabric architecture with r=4 (see Figure 1) that consists of three stages. The first and last stages comprise *r* Bandwidth-Variable Wavelength-Converting Switch (BV-WBCSs), represented as $I_{i}$ and $O_{j}$ (where 1⩽i, j⩽r), respectively. Only one Bandwidth-Variable Wavelength-Selective Space Switch (BV-WSSS) with *r* inputs and *r* outputs constitutes the center stage. There are *r* interstage links between every stage. WSW1(n,r,k) routes the m-slot connections, i.e., the connections that may use *m* subsequent FSUs (where one FSU represents 12.5 GHz of spectrum). BV-WBCS operates on m-slot connections and alters the spectrum of an m-slot connection from the spectrum of *m* FSUs at the input of BV-WBCS to another spectrum of *m* FSUs at the output.

The role of BV-WSSS is to switch an m-slot connection in the space domain without spectrum conversion capability. In [25], the internal structures of BV-WBCS and BV-WSSS are shown. The WSW1(r,n,k) inputs and outputs are numbered from 1 to *r*. In the same way, $I_{i}$ and $O_{j}$ are indexed. FSUs in the input fibers entering $I_{i}$ and in the output fibers leaving $O_{j}$ are numbered from 1 to *n*, while FSUs in the interstage links are numbered from 1 to *k*.

### 2.2. A New Connection Processing

The WSW1(r,n,k) switching fabric operates in two domains: space and frequency. Let us assume that WSW1(r,n,k) is nonblocking in the space domain, where BV-WSSS is a nonblocking switching fabric. Taking into account the state of the switching fabric shown in Figure 1, it should be noted that all m-slot connections that exist at the input of each $I_{i}$ are marked by the same pattern, while m-slot connections that appear at the output of each $O_{j}$ are distinguished by the same color. There is only one connection path between an input and an output of WSW1(r,n,k) there—a set of interstage links and a set of switching elements, such as $I_{i}$, $O_{j}$, and BV-WBSSS. With the exception of BV-WBSSS, all of these paths are disjoint, making WSW1(r,n,k) the nonblocking switching fabric in the space domain. However, if we consider the location of a particular m-slot connection in the frequency domain, we can locate connections that block each other. An *m*-slot connection from the input switch $I_{i}$ that uses FSUs numbered *x* to x+m−1 to the output switch $O_{j}$ with assigned slots *y* to y+m−1 is indicated by $\left(I_{i}\left[x\right],O_{j}\left[y\right],m\right)$. When frequency slot indices are not important, we will use the notation $\left(I_{i},O_{j},m\right)$; when we want to indicate only the switches that participate in a connection, we use the notation $\left(I_{i},O_{j}\right)$.

For example, the connection that demands slot 6 at input 1 blocks the connection that occupies slots 4–6 at input 3, since both connections are directed to $O_{2}$ (i.e., they use the same slot in the link between BV-WBSSS and $O_{2}$). The red connections at inputs 1 and 2 (which occupy slots 7 and 8) are also connected to the same output switch $O_{4}$, so they block each other. $I_{i}$ and $O_{j}$ eliminate the blocking situations in WSW1(r,n,k), i.e., connections in the blocking state are relocated to different frequency slots. We should mention that the operation of elastic optical networks imposes a connection $\left(I_{i},O_{j}\right)$, which must use adjacent FSUs. This operation is shown in Figure 1. $I_{i}$ aggregates all connections from input *i* to output *j* in the links between $I_{i}$ and BV-WBSSS (and between BV-WBSSS and $O_{j}$) to the set of connections indicated as $h_{i,j}$. The function of $O_{j}$ is to move connections that correspond to a specific $h_{i,j}$ to the requested position at the output of WSW1(r,n,k).

In this work, we employ the simultaneous connection model, that is, a model in which requests arrive at the input of the system at the same time and a set of compatible connections is set up simultaneously. Two connections are compatible if they use disjoint sets of FSUs in the input and/or output links. For example, the connections $\left(I_{1}\left[1\right],O_{2}\left[5\right],1\right)$ and $\left(I_{3}\left[4\right],O_{2}\left[4\right],3\right)$ are not compatible, since they both request slot 5 at output port 2. We assume that all FSUs in the input and output links are occupied by connections; that is, we have the maximum set of compatible connections. Although such a situation in real systems is uncommon, we can study all connection permutations in our analysis by including dummy connections, i.e., connections that use all free FSUs in the input and output links.

## 3. Control Algorithms

### 3.1. State Matrix

Let the maximum set of compatible connections be represented by C, and $X_{i,j}$ be a set of all connections $\left(I_{i},O_{j}\right)$ in C. In the considered switching fabric, the first and the last sections contain wavelength converters. This helps to simplify the analysis of the state of WSW1(r,n,k), since it is not necessary to analyze all the connections established in WSW1(r,n,k) but only the sum of the bandwidth units occupied between any pair ($I_{i}$, $O_{j}$). That is, we need to find relationships between the elements of the Cartesian product $I_{i}\times O_{j}$, which can be easily represented in the form of a matrix r×r. This matrix, denoted by $H^{r\times r}$, is defined in the following way [15],
(1)$$H^{r\times r}=\left[h_{i,j}\right],\;1⩽i,j⩽r,$$
where
(2)$$h_{i,j}=\sum_{X_{i,j}}m,\;\;\;\;X_{i,j}\subseteq C,\;\;\;\;\mathrm{and}\;\;\;\;X_{i,j}=\left\{\left(I_{i}\left[x\right],O_{i}\left[y\right],m\right)\right\}.$$

According to (Equation 2), element $h_{i,j}$ represents the aggregated spectrum of all connections $\left(I_{i},O_{j}\right)$. The matrix property is as follows,
(3)$$\sum_{i=1}^{r}h_{i,j}=\sum_{j=1}^{r}h_{i,j}=n,$$
i.e., in every individual input/output link, there are *n* FSUs. $H^{4\times 4}$ represents C for WSW1(4,n,k) switching fabrics, and
(4)$$H^{4\times 4}=\left[\begin{array}{cccc} h_{1,1} & h_{1,2} & h_{1,3} & h_{1,4}\\ h_{2,1} & h_{2,2} & h_{2,3} & h_{2,4}\\ h_{3,1} & h_{3,2} & h_{3,3} & h_{3,4}\\ h_{4,1} & h_{4,2} & h_{4,3} & h_{4,4} \end{array}\right].$$

In matrix (Equation 4), according to property (Equation 3), we have $h_{1,1}$ = *n* − ($h_{1,2}$ + $h_{1,3}$ + $h_{1,4}$) and $h_{1,1}$ = *n* − ($h_{2,1}$ + $h_{3,1}$ + $h_{4,1}$) and so on. This property of $H^{4\times 4}$ is later used in this paper to demonstrate the rearrangeable non-blocking (RNB) conditions of WSW1(r,n,k).

### 3.2. FSUs Assignment Algorithms for WSW1(4, n, r) Switching Fabrics

The objective of this section is to describe how FSUs should be assigned to the connections in the WSW1(4,n,k) switching fabrics. In a space-division switching fabric, a connection path, i.e., a set of resources used by a given connection, is composed of swithching elements (SEs) and interstage links (ILs) used by the connection. When two connections must use the same IL at the same time, one of these connections is blocked. In an EON switching fabric there is another dimension, wavelength. In this case, the connection path is determined not only by a set of SEs and ILs, but also by a set of FSUs on a given IL. To find two blocking connections, it is necessary to determine not only whether the connection paths of these two connections use the same IL(ILs), but also whether they use at least one common FSU.

The process of assigning FSUs to connection paths so that they do not block each other can be implemented in a way similar to that proposed in [15]. However, in the considered WSW1(4,n,k) switching fabric, the problem is much more complex, due to the larger number of inputs and outputs, resulting in a larger number of possible permutations. We propose six distinct FSUs assignment algorithms, denoted as AD1, AD2, AD3, AD4, AD5, and AD6. Then, we try to find the best one, or a group of them, for which the number *k* of required FSUs in ILs will be the lowest, yet allowing the realization of all possible sets of compatible connections [33].

The main idea of all proposed algorithms is to divide all FSUs in ILs into two subsets, in which we will establish connections corresponding to different $h_{i,j}$. Of course, the wavelength intervals of FSUs used for the connections represented by $h_{i,j}$ at the input port may be different from those assigned to these connections in ILs. The wavelength shift is the task of BV-WBCSs, from which the outermost sections are constructed. To determine the conditions under which all possible C can be realized in WSW1(r,n,k) (that is, it is RNB, as shown in [33]), we have to find the required value of *k* in each IL as the maximum value of the sum of the number of FSUs in both subsets.

To proceed with the description of the assignment algorithms, we have to add an important assumption about $H^{4\times 4}$. Each set of compatible connections in WSW1(4,n,k) is represented by a different state of the matrix $H^{4\times 4}$. We can reduce the number of states considered, assuming that $h_{1,1}$ is equal to or greater than the rest of the elements in $H^{4\times 4}$. Similarly, $h_{2,2}$ is equal to or greater than the other elements, except for those in row 1 and column 1 [34]. Finally, $h_{3,3}$ is equal to or greater than all the remaining elements in $H^{4\times 4}$ from rows 1 and 2 and elements from column 1 and 2. This property can be written as
(5)$$\begin{array}{c} h_{l,l}⩾h_{i,j},\;\mathrm{where}\;i,j=l,l+1,\ldots ,4. \end{array}$$

This assumption does not change the FSUs assignment process or the final result, but simplifies the explanation of the algorithms, their structure, and operation. If the criteria in (Equation 5) are not satisfied, the input and/or output stages, represented as rows and/or columns in the matrix $H^{4\times 4}$, should be modified by rearranging. For example, if the element $h_{i,j}$ is greater than $h_{1,1}$, $I_{1}$ is replaced with $I_{i}$ and $O_{1}$ with $O_{j}$, which makes $h_{i,j}$ become $h_{1,1}$.

To make it possible for each algorithm to assign FSUs to connections, we introduce the concept of subregions and critical pairs. The rows and columns of the matrix $H^{r\times r}$ represent the inputs and outputs of WSW1(r,n,k), respectively. Two elements located in the same row/column represent connections from/to the same input/output, and cannot occupy the same FSUs in ILs. Otherwise, the connections represented by one of these elements are blocked. For example, the connections represented by $h_{1,1}$ and $h_{2,2}$ may use the same set of FSUs since they come from and go to different outer stage switches. We call such elements “matched”. On the other hand, the connections represented by $h_{1,1}$ and $h_{2,1}$ have to use different FSUs, as they are directed to the same output switch, and we call them “mismatched”. Let us divide the elements of each row and each column into two pairs of elements, creating four groups of four elements each.

**Definition 1.** 
*Let $H^{4\times 4}$ represent C in WSW1(4,n,k) switching fabric working under algorithm ADx, where x=1,2,…,6. Each algorithm divides $H^{r\times r}$ into four disjoint subsets called quarters (denoted as $Q_{j}^{\mathrm{ADx}}$, where j=1,2,3,4). Each $Q_{j}^{\mathrm{ADx}}$ consists of four elements $h_{i,j}$, which form two pairs of matched elements.*


The way each algorithm assigns elements $h_{i,j}$ to quarters is shown in Figure 2. In each assignment, we can find pairs of $Q_{j}^{\mathrm{ADx}}$’s that are not in the same columns or rows. We surround these pairs with the same type of line and the same color, and the corresponding connections can occupy the same set of FSUs in ILs (elements matched). In contrast, $Q_{x}$’s surrounded by different line types always share the same rows or columns, i.e., they are mismatched (having to occupy disjoint sets of FSUs in ILs). For instance, in AD1  the quarters $Q_{1}^{\mathrm{AD}1}$ and $Q_{4}^{\mathrm{AD}1}$ contain $h_{1,1}$, $h_{1,2}$, $h_{2,1}$, $h_{2,2}$ and $h_{3,3}$, $h_{3,4}$, $h_{4,3}$, and $h_{4,4}$, respectively; these quarters are matched. In $Q_{1}^{\mathrm{AD}1}$, elements $h_{1,1}$ and $h_{2,2}$, as well as $h_{1,2}$ and $h_{2,1}$, are matched, while $h_{1,1}$ and $h_{1,2}$, as well as $h_{2,1}$ and $h_{2,2}$, or $h_{1,1}$ and $h_{2,1}$, as well as $h_{1,2}$ and $h_{2,2}$, are mismatched. When assigning slots, there will be no conflict between connections when mismatched quarters use disjoint sets of FSUs. Such mismatched quarters will be called “subregions”.

**Definition 2.** 
*Two mismatched quarters obtained in $H^{4\times 4}$ by the algorithm ADx are called subregions. The subregions are denoted as *ADx*.j, where j=1,2,3,4 and x=1,2,3,4,5,6.*


The divisions of $H^{4\times 4}$ elements into subregions for algorithms AD1, AD2, and AD6 are shown in Figure 3. Each of the four rectangles surrounding the matrix $H^{4\times 4}$ consists of a quarter surrounded by a solid line and a quarter surrounded by a dashed line (see Figure 3). When we have subregions determined by ADx, we have to divide the set of FSUs in ILs into two subsets, $S_{1}\left(\mathrm{ADx},C\right)$ and $S_{2}\left(\mathrm{ADx},C\right)$; slots in each subset will be used for connections represented in different quarters in subregions. As a result, the total number of FSUs in ILs for algorithm ADx, denoted by $k^{4\times 4}\left(\mathrm{ADx},C\right)$ is determined by the sum of slots needed in $S_{1}\left(\mathrm{ADx},C\right)$ and $S_{2}\left(\mathrm{ADx},C\right)$, that is,
(6)$$k^{4\times 4}\left(\mathrm{ADx},C\right)=|S_{1}\left(\mathrm{ADx},C\right)|+|S_{2}\left(\mathrm{ADx},C\right)|.$$

Any C can be realized by using ADx when we maximize (Equation 6) through all Cs, that is, when
(7)$$k^{4\times 4}\left(\mathrm{ADx}\right)=\max_{C}\left\{k^{4\times 4}\left(\mathrm{ADx},C\right)\right\}=\max_{C}\left\{\right|S_{1}\left(\mathrm{ADx},C\right)|+|S_{2}\left(\mathrm{ADx},C\right)\left|\right\}.$$

The connections shown in Figure 1 are set by using AD1, and the set of FSUs in each IL is divided into subsets $S_{1}\left(\mathrm{AD}1,C\right)$ and $S_{2}\left(\mathrm{AD}1,C\right)$ by bold vertical lines.

As we have $H^{4\times 4}$ divided into subregions, we will now show how the algorithms assign FSUs to connections. We discuss here only AD1 to limit the length of the paper; the assignment procedure for the other algorithms can be easily derived by analogy. First, we consider the connections represented by elements in $Q_{1}^{\mathrm{AD}1}$ and $Q_{4}^{\mathrm{AD}1}$, and the FSUs assigned to these connections will form the set $S_{1}\left(\mathrm{AD}1,C\right)$. The connections in $h_{1,1}$ and $h_{2,2}$ are matched, so they can use the same set of FSUs in ILs from $I_{1}$ and $I_{2}$ numbered from 1 to max{$h_{1,1}$;$h_{2,2}$}=$h_{1,1}$. Similarly, the connections in $h_{3,3}$ and $h_{4,4}$ are matched, so they can use the same set of FSUs in ILs from $I_{3}$ and $I_{4}$ and are numbered from 1 to max{$h_{3,3}$;$h_{4,4}$}=$h_{3,3}$. The connections in $h_{1,2}$ and $h_{2,1}$ are also matched, but are mismatched with connections in $h_{1,1}$ and $h_{2,2}$; therefore, they can use FSUs from $h_{1,1}$+1 to $h_{1,1}$+$h_{1,2}$ and from $h_{1,1}$+1 to $h_{1,1}$+$h_{2,1}$. The same approach can be used for the connections in $h_{3,4}$ and $h_{4,3}$—that is, they can use FSUs from $h_{3,3}$+1 to $h_{3,3}$+$h_{3,4}$ and from $h_{3,3}$+1 to $h_{3,3}$+$h_{4,3}$. As a result, we have already assigned *a* FSUs and a= $|S_{1}\left(\mathrm{AD}1,C\right)|$ =max{$h_{1,1}$+max{$h_{1,2}$; $h_{2,1}$};$h_{3,3}$+max{$h_{3,4}$; $h_{4,3}$}}; from property (Equation 3) $h_{1,1}⩾h_{2,2}$ and $h_{3,3}⩾h_{4,4}$. Next, we have to consider the connections represented by elements in $Q_{2}^{\mathrm{AD}1}$ and $Q_{3}^{\mathrm{AD}1}$, and the FSUs assigned to these connections will form the set $S_{2}\left(\mathrm{AD}1,C\right)$. By analogy, this set will contain FSUs numbered from a+1 to max{b;c}; where b=a+max{$h_{1,3}$; $h_{2,4}$} and c=a+max{$h_{3,1}$; $h_{4,2}$}.

The AD1 is listed as Algorithm 1, and the detailed assignment of FSUs is shown in Table 1. Similar procedures and tables can be obtained for algorithms from AD2 to AD6 by comparing $Q_{j}^{\mathrm{ADx}}$ presented in Figure 2b–f with that obtained for AD1 (Figure 2a), and by comparing how FSUs are assigned to elements of $Q_{j}^{\mathrm{AD}1}$ in Table 1.
**Algorithm 1:** (AD1)**Data**: C**Result**: FSUs assigned to connections in C**1** Create matrix $H^{4\times 4}$
**2** Assign FSUs to the connections according to Table 1

### 3.3. The Maximum Number of FSUs in the Algorithms

We now determine the maximum number of FSUsto be occupied by each pair of mismatched quarters; that is, we derive the maximum values for (Equation 7) for each algorithm. The number of FSUs required by each $S_{1}\left(\mathrm{ADx},C\right)$ and each $S_{2}\left(\mathrm{ADx},C\right)$ depends on the elements of $H^{4\times 4}$ surrounded by solid and dashed lines, respectively. Similarly, as in the description of the algorithms, we provide a detailed analysis for AD1 and present the final equations for the remaining algorithms. From Table 1, we obtain
(8)$$|S_{1}\left(\mathrm{ADx},C\right)|=\max_{C}\left\{\max\left\{h_{1,1};h_{2,2}\right\}+\max\left\{h_{1,2};h_{2,1}\right\};\max\left\{h_{3,3};h_{4,4}\right\}+\max\left\{h_{3,4};h_{4,3}\right\}\right\}$$
and
(9)$$|S_{2}\left(\mathrm{ADx},C\right)|=\max_{C}\left\{\max\left\{h_{1,3};h_{2,4}\right\}+\max\left\{h_{2,3};h_{1,4}\right\};\max\left\{h_{3,1};h_{4,2}\right\}+\max\left\{h_{3,2};h_{4,1}\right\}\right\}.$$

In general, each element of each pair in (Equation 8) and (Equation 9) can be greater. Therefore, we must consider all the cases to determine the maximum needed value of *k*. When the first sums in (Equation 8) and (Equation 9) are greater, we choose the maximum between pairs ($h_{1,1}$;$h_{2,2}$), ($h_{1,2}$;$h_{2,1}$), ($h_{1,3}$;$h_{2,4}$), and ($h_{2,3}$;$h_{4,1}$). The elements in these pairs form the subregion AD1.1 (see Figure 3a). In the subregions ADx.1 and ADx.4, quarters are placed horizontally (H), while in ADx.2 and ADx.3, quarters are placed vertically (V). The elements with maximum values in the subregions can be located in four scenarios: HH, HV, VH, and VV (see Figure 4), while in each scenario we have four cases denoted by (a), (b), (c), and (d). These cases for scenario HH in AD1.1 are presented in Figure 5.

In the equations listed in Appendix A, we provide the results for the possible maximum values of Equation (Equation 7) for all the scenarios and cases for AD1. When creating these formulas, we take into account the properties (Equation 3) and (Equation 5). Here we explain two cases. In the AD1.1 scenario HH and case (a) $h_{1,1}$, $h_{1,2}$, $h_{1,3}$, and $h_{1,4}$ are the maximum values. This means that $\max_{C}\left\{k^{4\times 4}\left(\mathrm{AD}1.1;C\right)\right\}=h_{1,1}+h_{1,2}+h_{1,3}+h_{1,4}$, and this sum is equal to *n* (Equation 3). In the same scenario, but in case (c), we have $|S_{1}\left(\mathrm{ADx},C\right)|=h_{2,1}+h_{2,2}$ and $|S_{2}\left(\mathrm{ADx},C\right)|=h_{1,3}+h_{1,4}$. The maximum value of $h_{2,1}+h_{2,2}$ can be *n*. However, since $h_{1,1}⩾h_{2,2}$ we get h1,1⩾⌈n2⌉ and h2,2⩽⌊n2⌋ Since h1,1⩾⌈n2⌉, the maximum value of $h_{1,3}$ + $h_{1,4}$ = *n* − $h_{1,1}$, i.e., h1,3+h1,4=⌊n2⌋. As the result, $k^{4\times 4}\left(\mathrm{AD}1.1;C\right)$ = n+⌊n2⌋.

The listed equations give the maximum values of *k* for a given subregion, scenario, and case. Now, we have to find for which of these equations *k* reaches the highest value through all possible Cs, that is
(10)$$k^{4\times 4}\left(\mathrm{AD}1\right)=\max_{C}\left\{k^{4\times 4}\left(\mathrm{AD}1.1;C\right);k^{4\times 4}\left(\mathrm{AD}1.2;C\right);k^{4\times 4}\left(\mathrm{AD}1.3;C\right);k^{4\times 4}\left(\mathrm{AD}1.4;C\right)\right\}.$$

When we look at Table 2, where we included the results of all subregions, scenarios, and cases for algorithms AD1, AD2, and AD6, we can conclude that *n* ≤ $k^{4\times 4}\left(\mathrm{AD}1\right)$ ≤ 2n. This means that there are Cs for which we can realize all connections in less than 2n slots, but there are also cases where C requires 2n slots. However, when we need 2n slots for C in the algorithm AD1, we may need less than 2n slots when using another algorithm, for example AD2 or AD6. The results of these algorithms are also included in Table 2. We omit AD3 and AD5, because the arrangement of elements in AD2 is similar to the transpose of AD3, and the arrangement of elements in AD6 is similar to the transpose of AD5; moreover, the results for these algorithms are exactly the same as for AD2 and AD6, respectively. The problem of finding the best value of *k* when all the algorithms are used is the subject of the next section.

## 4. The New Algorithm AD7 and RNB Conditions

The results of allocating FSUs to connections by algorithms AD1–AD6 vary widely. The question is which algorithm should be used to minimize the number of required FSUs in ILs, yet make it possible to realize any C. This minimum value of *k* makes the WSW1(4,n,k) rearrangeable and is given by the following formula: (11)$$\begin{array}{c} k_{\mathrm{RNB}}^{4\times 4}⩾\max_{C}\left\{\min\left\{k^{4\times 4}\left(\mathrm{AD}1;C\right);k^{4\times 4}\left(\mathrm{AD}2;C\right);k^{4\times 4}\left(\mathrm{AD}3;C\right);k^{4\times 4}\left(\mathrm{AD}4;C\right);k^{4\times 4}\left(\mathrm{AD}5;C\right);k^{4\times 4}\left(\mathrm{AD}6;C\right)\right\}\right\} \end{array}.$$

As stated in the previous section, we have $k^{4\times 4}\left(\mathrm{AD}2;C\right)=k^{4\times 4}\left(\mathrm{AD}3;C\right)$ and $k^{4\times 4}\left(\mathrm{AD}5;C\right)=k^{4\times 4}\left(\mathrm{AD}6;C\right)$; therefore, we can exclude AD3 and AD5 from (Equation 11). When we compared the results for AD1, AD2, and AD6 (see Table 2) with the results for AD4, we concluded that for any C we had $\min\left\{k^{4\times 4}\left(\mathrm{AD}1;C\right);k^{4\times 4}\left(\mathrm{AD}2;C\right);k^{4\times 4}\left(\mathrm{AD}6;C\right)\right\}⩽k^{4\times 4}\left(\mathrm{AD}4;C\right)$; therefore, we could also exclude AD4 from (Equation 11). Consequently, we finally obtain
(12)$$\begin{array}{c} k_{\mathrm{RNB}}^{4\times 4}=k^{4\times 4}\left(\mathrm{AD}7\right)=\max_{C}\left\{\min\left\{k^{4\times 4}\left(\mathrm{AD}1;C\right);k^{4\times 4}\left(\mathrm{AD}2;C\right);k^{4\times 4}\left(\mathrm{AD}6;C\right)\right\}\right\}, \end{array}$$
where AD7 is given as Algorithm 2.
**Algorithm 2:** (AD7)
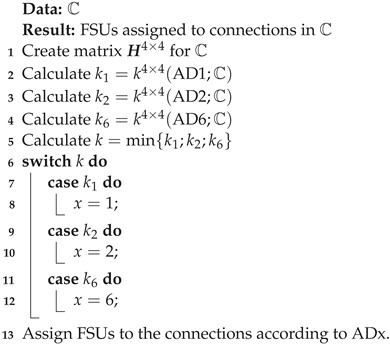


We explain the idea of AD7 performance by means of the example presented in Figure 6. This example shows two matrices $H^{4\times 4}$ representing connections in two switching fabrics, WSW1(4,4,k) and WSW1(4,5,k) (see Figure 6a,b). In Figure 6a, we have n=4 (*n* is even) and when we use AD1 to assign FSUs, we need $k^{4\times 4}\left(\mathrm{AD}1,C\right)=2n=8$ slots. This maximum value is obtained for the subregion AD1.3, scenario VV, and case (c) (see Table 2) for which $k=h_{1,4}+h_{2,4}+h_{3,3}+h_{4,3}=2+2+2+2=8$. However, when we use AD2 for the same subregion, scenario and case, we see that these connections can be realized by using only $k=h_{2,4}+h_{3,4}+h_{1,3}+h_{4,3}=2+0+0+2=4$ slots. For AD6, we also have $k=h_{2,4}+h_{4,4}+h_{1,3}+h_{3,3}=2+0+0+2=4$ slots. However, this number of slots is not sufficient to realize connections in other the subregions by AD2 or AD6. Both algorithms need 6 FSUs for subregions AD2.2 and AD6.2. On the other hand, there may be other connection patterns, for which AD2 and AD6 need more FSUs. We see this in Table 2 since for AD2.3, VV scenario and case (c), the maximum number of required *k* is 2n but for AD6.3—only n+⌊n2⌋. Similarly, we can analyze Figure 6b, where we have n=5 (*n* is odd). For AD1, we need k=9 FSUs, while AD2 and AD6 can fit connections in k=5 FSUs. The role of AD7 is to determine, for a given C, which of the algorithms AD1, AD2, or AD6 requires the fewest number of FSUs in ILs, and to use this algorithm for assigning slots to connections.

The question now is what the minimum number of required FSUs in ILs is sufficient to ensure that AD7 always ends with success for any C. The answer to this question is given in the following theorem.

**Theorem 1.** 
*The WSW1(4,n,k) switching fabric is RNB for m-slot connections, 1⩽m⩽n, and when n⩾4 under the algorithm AD7 if*

(13)
k⩾kRNB4×4=n+⌊2n3⌋.



**Proof.** In AD7, for a given C, we check for which of the algorithms AD1, AD2, or AD6 we need the lowest number of FSUs. Then, these minimum values calculated for each C must be maximized (see (Equation 12)). We can find the better algorithm by calculating the number of FSUs for each algorithm in each subregion for each scenario and case. These values are shown in Table 3, which we obtained from Table 2. We see that the maximum value in this table is n+⌊2n3⌋; therefore, this number of FSUs is sufficient to realize all possible compatible connection patterns. □

## 5. Comparisons and Numerical Results

Algorithm AD7 gives the best results of all the algorithms proposed in this paper. Now, we compare this algorithm with the algorithms CA6 and CA7 proposed in [15]. We first consider the WSW1(4,n,k) switching fabric. The values of $k_{\mathrm{RNB}}^{4\times 4}$ are compared in Table 4 and plotted in Figure 7. As for the CA6, AD7 allows reducing the number of FSUs in ILs by more than 16%. In case of CA7, this reduction is greater than 40%. For instance, when n=160 (a typical value for band C in optical fibers when frequency slot is 25 GHz wide), CA6 requires 320 FSUs, CA7—448 FSUs, while AD7 finishes with success when there are only 266 FSUs in ILs. This is illustrated in Table 4 and Figure 7, where kRNBr×r(CA6)=⌈r2⌉n and kRNBr×r(CA7)=⌈r3⌉(n+⌊2n5⌋) [15]. Those previously developed algorithms (CA6, CA7) are based on the analysis of r=2 and r=3; however, AD7 is developed based on the analysis of r=4, which delivers the most optimal and efficient result represented by the lowest number of FSUs in ILs. In addition, both algorithms CA6 and CA7 are multiplied by the factor of ⌈r2⌉, and ⌈r3⌉, respectively, which results in multiplication by 2 in case of r=4. That would affect the overall result to be multiplied by a factor of 2, which eventually leads to an increase of the number of FSUs in ILs. Hence, $k_{\mathrm{RNB}}^{4\times 4}\left(CA6\right)=2n$ and kRNB4×4(CA7)=2(n+⌊2n5⌋), which means that both algorithms give greater value than kRNB4×4(AD7)=n+⌊2n3⌋. Therefore, we notice that AD7, when applied for $H^{4\times 4}$, yields a better result than previously published algorithms. For instance, $k_{\mathrm{RNB}}^{4\times 4}\left(\mathrm{AD}7\right)$ = 8, where $k_{\mathrm{RNB}}^{4\times 4}\left(CA6\right)$ = 10 and $k_{\mathrm{RNB}}^{4\times 4}\left(CA7\right)$ = 14.

When r>4, we compare some results for r= 8, 16 and 32 and selected values of *n*. FSUs are assigned to connections by dividing $H^{r\times r}$ into submatrices $H^{4\times 4}$, similar to what is proposed in [15], and by using AD7 for each submatrix. The number of FSUs in ILs is calculated in this case by the following equation:(14)kRNBr×r(AD8)⩾kr×r(AD8)=⌈r4⌉×kRNB4×4(AD7).

The results are compared in Table 5 with $k_{\mathrm{RNB}}^{r\times r}\left(CA6\right)$ and $k_{\mathrm{RNB}}^{r\times r}\left(CA7\right)$ [15], where kRNBr×r(CA6)=⌈r2⌉n and kRNBr×r(CA7)=⌈r3⌉(n+⌊2n5⌋). We can see that AD7, when employed for $k_{\mathrm{RNB}}^{r\times r}$, outperforms all the algorithms published. For example, when n=20 we get $k_{\mathrm{RNB}}^{8\times 8}\left(CA6\right)=80$ and $k_{\mathrm{RNB}}^{8\times 8}\left(CA7\right)=84$, where $k_{\mathrm{RNB}}^{8\times 8}\left(\mathrm{AD}8\right)=66$. This shows how AD8 yields better results when compared with other previously known algorithms.

In addition, as we illustrated previously, CA6 and CA7 are developed based on the analysis of $H^{2\times 2}$ and $H^{3\times 3}$, respectively, where AD7 is developed based on the analysis of $H^{4\times 4}$. Therefore, when it comes to finding the number of FSUs in ILs, each algorithm is multiplied by a factor representing the value of *r*, i.e., algorithm CA6 is multiplied by a factor of ⌈r2⌉, algorithm CA6 is multiplied by a factor of ⌈r3⌉, and algorithm AD7 is multiplied by a factor of ⌈r4⌉ donated as AD8. In case of r=8, the factor for CA6, CA7, and AD8, respectively, will be (4, 3, and 2); similarly for r=16 the factor will be (8, 6, and 4), and finally, for r=32 the factor will be (16, 11, and 8). Accordingly, as *r* increases, the factor difference between algorithms increases, which can cause CA7 to have a lower number of FSUs in ILs than CA6 (see r=32 in Table 5).

We compared the proposed algorithm result with other results presented in [15]. We also investigated the results presented in [27] and found that they were not better than that presented in [15]. For instance, when $k_{\mathrm{RNB}}^{r\times r}\left(CA6\right)=20$, $k_{\mathrm{RNB}}^{r\times r}\left(CA7\right)=28$, and $k_{\mathrm{RNB}}^{r\times r}\left(AD7\right)=16$, the algorithms in [27] achieves $k_{\mathrm{SNB}}^{r\times r}=124$, and $k_{\mathrm{RNB}}^{r\times r}=40$. Furthermore, SNB derived in [13] requires even more FSUs than that presented in [15], as $k_{\mathrm{SNB}}^{8\times 8}=225$, where $k_{\mathrm{RNB}}^{8\times 8}\left(AD7\right)=132$. Similarly, the WNB proposed in [35] is not better than [15]. Therefore, it is sufficient to compare result for the proposed algorithms to the best-known result to date.

Finally, we should mention that the algorithm is very simple. The complexity of the algorithm is determined mainly by matrix preparation and rearranging such that the property (Equation 3) is fulfilled. When the matrix $H^{4\times 4}$ is given, AD7 chooses the final algorithm to assign FSUs to connections in O(1) time, and then the selected algorithm assigns slots by using a fixed table. For AD1 this assignment is given in Table 1.

## 6. Conclusions

In this paper, we proposed seven algorithms for assigning frequency slots to connections in WSW1(4,n,k) switching fabrics serving multislot connections. Algorithms AD1 to AD6 are based on matrix decomposition, while AD7 uses all the algorithms mentioned before. For a given connection pattern, it uses those algorithms that require the fewest frequency slots. This approach was previously proposed in [15] but was limited to switching fabrics with r=3. When r=4, the number of slot assignment patterns increases rapidly, and many more cases must be analyzed to find the number of frequency slots, which ensures that the algorithms always end with success. The number that makes WSW1(4,n,k) rearrangeably nonblocking is given in Theorem 1. Algorithm AD7 improves the previously known results by more than 16% or even 40%, depending on which algorithm is used. We also extrapolated the number of required frequency slots for WSW1(r,n,k) switching fabrics with r>4. In this case, the proposed algorithm also outperforms the previously known algorithms. It should be noted that the proposed solutions provide sufficient conditions for RNB. The necessity is not shown; this means that there is still place for other algorithms, which reduces the number of required frequency slots. However, we know that this number cannot be lower than n+⌊n4⌋ [15].

In this work, *k* is the only parameter we can use to obtain RNB conditions. When we have these conditions, we can consider how much power or how many elements the switching fabric needs. However, the number of tunable wavelength converters is rather determined by the switching fabric capacity and architecture (*r* switches in the input/output stages, one switch in the center stage). Therefore, *k* is more important since it determines the range of required converters. Moreover, the complexity of the algorithm is very low, since we have to calculate the result according to three simple algorithms from the algorithm AD8 (Equation 12), as shown in (2), and use that with the lowest result. Then FSUs are assigned to connections by using the assignment given for each algorithm, such as in Table 1. The most time-consuming task is to prepare the matrix from the set of connections and sort it in such a way that conditions (Equation 3) and (Equation 5) are fulfilled.

## Figures and Tables

**Figure 1 sensors-23-03615-f001:**
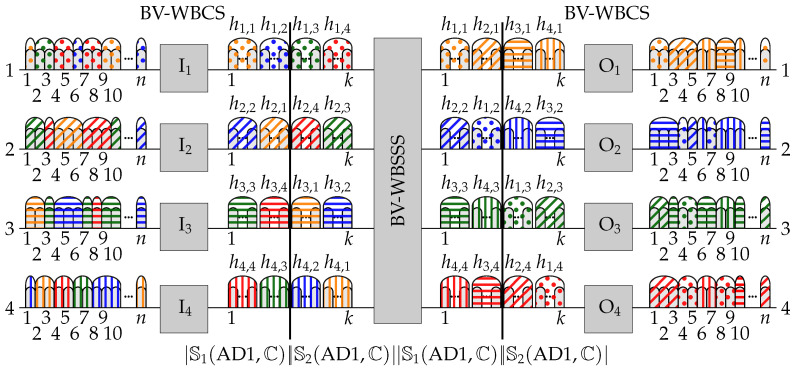
A general architecture of WSW1(r,n,k) and connections processing.

**Figure 2 sensors-23-03615-f002:**
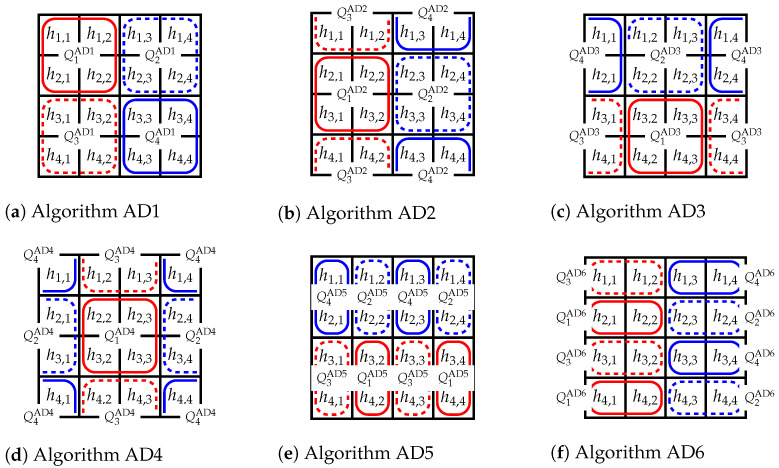
Division of the matrix $H^{4\times 4}$ into quarters in algorithms AD1–AD6.

**Figure 3 sensors-23-03615-f003:**
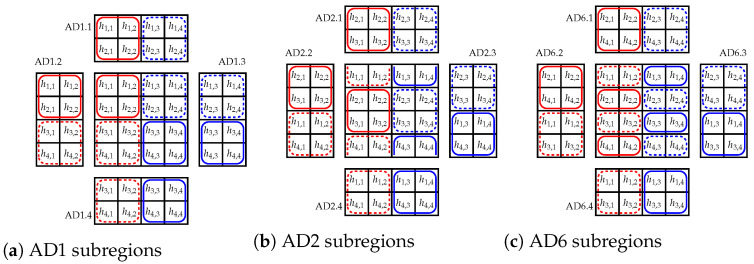
Division of $H^{4\times 4}$ into subregions by Algorithms AD1, AD2, and AD6.

**Figure 4 sensors-23-03615-f004:**
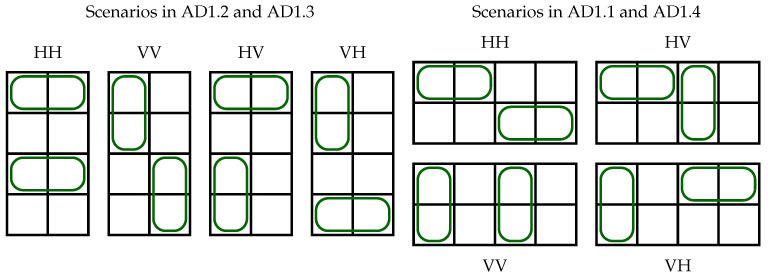
Location of elements with the maximum value in horizontally (H) and vertically (V) stacked quarters.

**Figure 5 sensors-23-03615-f005:**
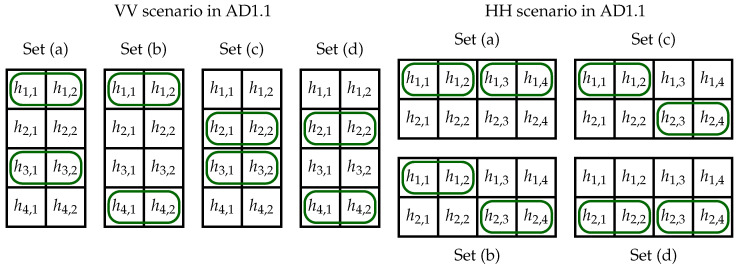
Four cases with maximum values of elements $h_{i,j}$ in VV and HH scenarios in AD1.1.

**Figure 6 sensors-23-03615-f006:**
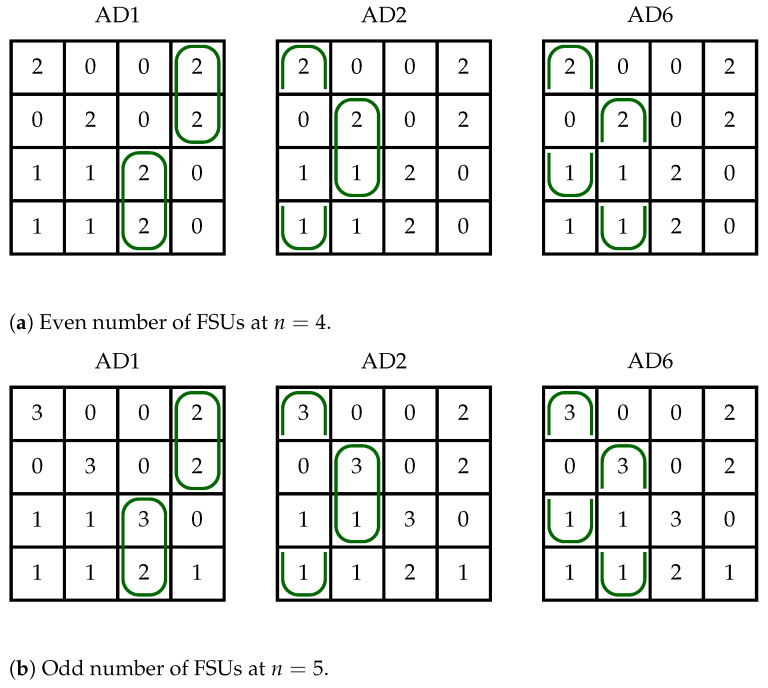
Example of applying AD2 and AD6 instead of AD1 to achieve lower value for *k*.

**Figure 7 sensors-23-03615-f007:**
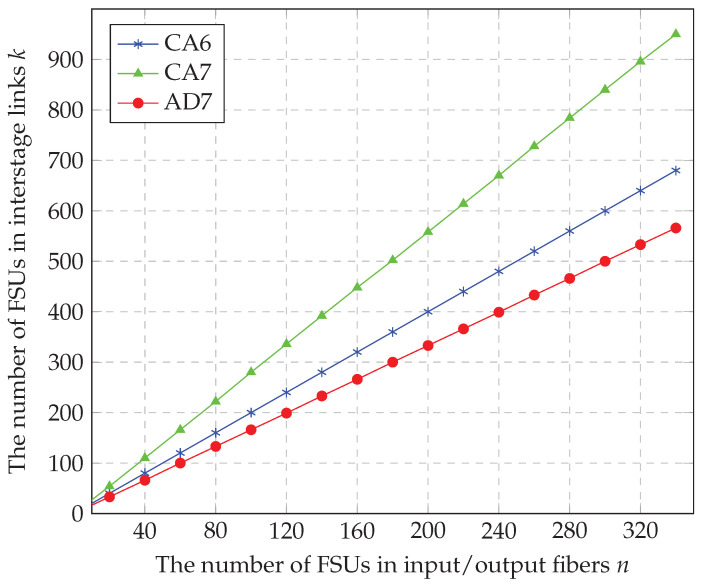
Comparison of $k_{\mathrm{RNB}}^{4\times 4}$ presented in [15] and required by Theorem 1.

**Table 1 sensors-23-03615-t001:** Assignment of FSUs to connections by AD1.

Algorithm AD1				
$h_{i,j}$	Index Numbers of Assigned FSUs		$Q_{j}^{\mathrm{AD}1}$	Set
	From	To		
$h_{1,1}$	1	$h_{1,1}$	$Q_{1}^{\mathrm{AD}1}$	$S_{1}\left(\mathrm{AD}1,C\right)$
$h_{2,2}$	1	$h_{2,2}$	$Q_{1}^{\mathrm{AD}1}$	
$h_{1,2}$	$h_{1,1}$ +1	$h_{1,1}$+$h_{1,2}$	$Q_{1}^{\mathrm{AD}1}$	
$h_{2,1}$	$h_{1,1}$ +1	$h_{1,1}$+$h_{2,1}$	$Q_{1}^{\mathrm{AD}1}$	
$h_{3,3}$	1	$h_{3,3}$	$Q_{4}^{\mathrm{AD}1}$	
$h_{4,4}$	1	$h_{4,4}$	$Q_{4}^{\mathrm{AD}1}$	
$h_{3,4}$	$h_{3,3}$ +1	$h_{3,3}$+$h_{3,4}$	$Q_{4}^{\mathrm{AD}1}$	
$h_{4,3}$	$h_{3,3}$ +1	$h_{3,3}$+$h_{4,3}$	$Q_{4}^{\mathrm{AD}1}$	
$h_{1,3}$	a+1	a+ $h_{1,3}$	$Q_{2}^{\mathrm{AD}1}$	$S_{2}\left(\mathrm{AD}1,C\right)$
$h_{2,4}$	a+1	a+ $h_{2,4}$	$Q_{2}^{\mathrm{AD}1}$	
$h_{1,4}$	b+1	b+ $h_{1,4}$	$Q_{2}^{\mathrm{AD}1}$	
$h_{2,3}$	b+1	b+ $h_{2,3}$	$Q_{2}^{\mathrm{AD}1}$	
$h_{3,1}$	a+1	a+ $h_{3,1}$	$Q_{3}^{\mathrm{AD}1}$	
$h_{4,2}$	a+1	a+ $h_{4,2}$	$Q_{3}^{\mathrm{AD}1}$	
$h_{3,2}$	c+1	c+ $h_{3,2}$	$Q_{3}^{\mathrm{AD}1}$	
$h_{4,1}$	c+1	c+ $h_{4,1}$	$Q_{3}^{\mathrm{AD}1}$	

a = max{$h_{1,1}$+max{$h_{1,2}$; $h_{2,1}$};$h_{3,3}$+max{$h_{3,4}$; $h_{4,3}$}};
b = a + max{$h_{1,3}$;
$h_{2,4}\}$;
c = a + max{$h_{3,1}$;
$h_{4,2}\}$.

**Table 2 sensors-23-03615-t002:** AD1 Algorithm scenarios comparison.

Subregions	Scenarios	Algorithm AD1			
		set(a)	set(b)	set(c)	set(d)
AD1.1	HH	*n*	*n*+⌊2n3⌋	*n*+⌊n2⌋	*n*
	VV	*n*+⌊n2⌋	*n*+⌊n2⌋	*n*+⌊n2⌋	*n*+⌊n2⌋
	VH	*n*+⌊n2⌋	*n*+⌊2n3⌋	*n*	*n*+⌊n2⌋
	HV	*n*+⌊n2⌋	*n*+⌊n2⌋	*n*+⌊n2⌋	*n*+⌊n2⌋
AD1.2	HH	*n*+⌊n2⌋	*n*+⌊n2⌋	*n*+⌊n2⌋	*n*+⌊n2⌋
	VV	*n*	*n*+⌊2n3⌋	*n*+⌊n2⌋	*n*
	VH	*n*+⌊n2⌋	*n*+⌊n2⌋	*n*+⌊n2⌋	*n*+⌊n2⌋
	HV	*n*+⌊n2⌋	*n*+⌊2n3⌋	*n*	*n*+⌊n2⌋
AD1.3	HH	*n*+⌊n2⌋	*n*+⌊n2⌋	*n*+⌊n2⌋	*n*+⌊n2⌋
	VV	*n*	*n*+⌊n2⌋	2n,2n−1	*n*
	VH	*n*+⌊n2⌋	*n*	*n*+⌊2n3⌋	*n*+⌊n2⌋
	HV	*n*+⌊n2⌋	*n*+⌊n2⌋	*n*+⌊n2⌋	*n*+⌊n2⌋
AD1.4	HH	*n*	*n*+⌊n2⌋	2n,2n−1	*n*
	VV	*n*+⌊n2⌋	*n*+⌊n2⌋	*n*+⌊n2⌋	*n*+⌊n2⌋
	VH	*n*+⌊n2⌋	*n*+⌊n2⌋	*n*+⌊n2⌋	*n*+⌊n2⌋
	HV	*n*+⌊n2⌋	*n*	*n*+⌊2n3⌋	*n*+⌊n2⌋
**Subregions**	**Scenarios**	**Algorithm AD2**			
		**set(a)**	**set(b)**	**set(c)**	**set(d)**
AD2.1	HH	*n*	2n	*n*+⌊n2⌋	*n*
	VV	*n*+⌊3n4⌋	2n	*n*+⌊n3⌋	*n*+⌊n2⌋
	VH	*n*+⌊n2⌋	2n	*n*	*n*+⌊n2⌋
	HV	*n*+⌊n2⌋	2n	*n*+⌊n2⌋	*n*+⌊n2⌋
AD2.2	HH	2n	*n*+⌊n2⌋	*n*+⌊n2⌋	⌊2n3⌋+⌊2n3⌋
	VV	*n*	2n	⌊2n3⌋+⌊2n3⌋	*n*
	VH	2n	*n*+⌊n2⌋	*n*+⌊n2⌋	⌊n2⌋+⌊2n3⌋
	HV	*n*+⌊n2⌋	2n	*n*	*n*+⌊n2⌋
AD2.3	HH	⌊2n3⌋+⌊2n3⌋	*n*+⌊n2⌋	*n*+⌊n2⌋	2n
	VV	*n*	*n*+⌊n2⌋	2n	*n*
	VH	*n*+⌊n4⌋	*n*+⌊n2⌋	*n*+⌊n2⌋	2n
	HV	*n*+⌊n2⌋	*n*+⌊n2⌋	2n	*n*+⌊n2⌋
AD2.4	HH	*n*	2n	⌊3n4⌋+⌊2n3⌋	*n*
	VV	*n*+⌊n2⌋	*n*+⌊n3⌋	2n	*n*+⌊3n4⌋
	VH	⌊n2⌋+⌊2n3⌋	*n*+⌊n2⌋	*n*+⌊n2⌋	2n
	HV	2n	*n*+⌊n2⌋	*n*+⌊n2⌋	*n*+⌊n4⌋
**Subregions**	**Scenarios**	**Algorithm AD3**			
		**set(a)**	**set(b)**	**set(c)**	**set(d)**
AD6.1	HH	*n*	2n	⌊3n4⌋+⌊2n3⌋	*n*
	VV	2n	*n*+⌊3n4⌋	*n*+⌊n2⌋	*n*+⌊n3⌋
	VH	*n*+⌊n2⌋	2n	*n*	*n*+⌊n2⌋
	HV	2n	*n*+⌊n2⌋	*n*+⌊n2⌋	*n*+⌊n4⌋
AD6.2	HH	2n	*n*+⌊n2⌋	*n*+⌊n2⌋	*n*+⌊n4⌋
	VV	*n*	2n	⌊2n3⌋+⌊2n3⌋	*n*
	VH	2n	*n*+⌊n2⌋	*n*+⌊n2⌋	⌊n2⌋+⌊2n3⌋
	HV	*n*+⌊n2⌋	2n	*n*	*n*+⌊n2⌋
AD6.3	HH	*n*+⌊n2⌋	*n*+⌊n2⌋	*n*+⌊n2⌋	2n
	VV	*n*	2n	*n*+⌊n2⌋	*n*
	VH	*n*+⌊n2⌋	2n	*n*+⌊n2⌋	*n*+⌊n2⌋
	HV	*n*+⌊n4⌋	*n*+⌊n2⌋	*n*+⌊n2⌋	2n
AD6.4	HH	*n*	2n	*n*+⌊n2⌋	*n*
	VV	*n*+⌊n3⌋	*n*+⌊n2⌋	*n*+⌊3n4⌋	2n
	VH	⌊n2⌋+⌊2n3⌋	*n*+⌊n2⌋	*n*+⌊n2⌋	2n
	HV	*n*+⌊n2⌋	2n	*n*+⌊n2⌋	*n*+⌊n2⌋

**Table 3 sensors-23-03615-t003:** Algorithm AD7 scenarios.

Algorithm		AD7			
Subregions	Scenarios	a	b	c	d
AD7.1	HH	*n*	*n*+⌊2n3⌋	⌊3n4⌋+⌊2n3⌋	*n*
	VV	*n*+⌊n2⌋	*n*+⌊n2⌋	*n*+⌊n3⌋	*n*+⌊n3⌋
	VH	*n*+⌊n2⌋	*n*+⌊2n3⌋	*n*	*n*+⌊n2⌋
	HV	*n*+⌊n2⌋	*n*+⌊n2⌋	*n*+⌊n2⌋	*n*+⌊n4⌋
AD7.2	HH	*n*+⌊n2⌋	*n*+⌊n2⌋	*n*+⌊n2⌋	*n*+⌊n4⌋
	VV	*n*	*n*+⌊2n3⌋	⌊2n3⌋+⌊2n3⌋	*n*
	VH	*n*+⌊n2⌋	*n*+⌊n2⌋	*n*+⌊n2⌋	⌊n2⌋+⌊2n3⌋
	HV	*n*+⌊n2⌋	*n*+⌊2n3⌋	*n*	*n*+⌊n2⌋
AD7.3	HH	⌊2n3⌋+⌊2n3⌋	*n*+⌊n2⌋	*n*+⌊n2⌋	*n*+⌊n2⌋
	VV	*n*	*n*+⌊n2⌋	*n*+⌊n2⌋	*n*
	VH	*n*+⌊n4⌋	*n*	*n*+⌊n2⌋	*n*+⌊n2⌋
	HV	*n*+⌊n4⌋	*n*+⌊n2⌋	*n*+⌊n2⌋	*n*+⌊n2⌋
AD7.4	HH	*n*	*n*+⌊n2⌋	⌊3n4⌋+⌊2n3⌋	*n*
	VV	*n*+⌊n3⌋	*n*+⌊n3⌋	*n*+⌊n2⌋	*n*+⌊n2⌋
	VH	⌊n2⌋+⌊2n3⌋	*n*+⌊n2⌋	*n*+⌊n2⌋	*n*+⌊n2⌋
	HV	*n*+⌊n2⌋	*n*	*n*+⌊n2⌋	*n*+⌊n4⌋

**Table 4 sensors-23-03615-t004:** Comparison of $k_{\mathrm{RNB}}^{4\times 4}$ presented in [15] and required by Theorem 1.

	$k_{\mathrm{RNB}}^{4\times 4}$				
*n*	CA6 [15]	CA7 [15]	AD7	AD7 to CA6 [15]	AD7 to CA7 [15]
5	10	14	8	20%	42.857%
10	20	28	16	20%	42.857%
15	30	42	25	16.7%	40.476%
20	40	56	33	17.5%	41.071%
40	80	112	66	17.5%	41.071%
60	120	168	100	16.7%	40.476%
80	160	224	133	16.875%	40.625%
160	320	448	266	16.875%	40.625%
320	640	896	533	16.7%	40.513%

**Table 5 sensors-23-03615-t005:** Comparison of $k_{\mathrm{RNB}}^{r\times r}$ presented in [15] and required by Theorem 1.

*n*	r=8			r=16			r=32		
	CA6	CA7	AD8	CA6	CA7	AD8	CA6	CA7	AD8
20	80	84	66	160	168	132	320	308	264
40	160	168	132	320	336	264	640	616	528
60	240	252	200	480	504	400	960	924	800
80	320	336	266	640	672	532	1280	1232	1064
100	400	420	332	800	840	664	1600	1540	1328
120	480	504	400	960	1008	800	1920	1848	1600
140	560	588	466	1120	1176	932	2240	2156	1864
160	640	672	532	1280	1344	1064	2560	2464	2128
180	720	756	600	1440	1512	1200	2880	2772	2400
200	800	840	666	1600	1680	1332	3200	3080	2664
220	880	924	732	1760	1848	1464	3520	3388	2928
240	960	1008	800	1920	2016	1600	3840	3696	3200
260	1040	1092	866	2080	2184	1732	4160	4004	3464
280	1120	1176	932	2240	2352	1864	4480	4312	3728
300	1200	1260	1000	2400	2520	2000	4800	4620	4000
320	1280	1344	1066	2560	2688	2132	5120	4928	4264

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
