# Peer review of "Simultaneous Connections Routing in Wavelength–Space–Wavelength Elastic Optical Switches"

_sensors, 2023, doi:10.3390/s23073615_

Round 1
Reviewer 1 Report
This article investigates new Algorithms for Simultaneous Connections Routing in Elastic Optical Networks. Switching fabrics of greater capacity with more than three inputs and outputs have been investigated, and the obtained results are used to estimate routing in the switching fabric with an arbitrary number of inputs and outputs. The topic is important and relevant to the research area of optical networks. Here are some notes to address:
The equations in pages 9 to 12 should be moved to the appendix.
The comparison results for different r in table 5 should be explained in more detail.
The reference list should be expanded to reflect related research in the field.
There are a few typo errors to fix.
Reviewer 2 Report
The authors of this manuscript propose an algorithm for assigning frequency slots to connections in wavelength-space-wavelength switching fabrics for multi-slot links. The notion of this paper is very interesting and is valuable for future elastic optical networks. However, some of my concerns are:
1) The authors say that they introduce 7 algorithms. However, the last algorithm outperforms all of the previous ones, thus rendering them inadequate for the specific task. This is highlighted in Section 5, which stats with the sentence: "Algorithm AD7 gives the best results of all algorithms proposed in this paper".
2) The authors only showcase the performance of the suggested scheme in one very simple figure. More results should be included comparing the proposed approach with other state of the art ones.
3) Only the "The number of FSUs in interstage links k" is used as a performance metric for the suggested system. More insightful metrics should be used such as complexity of the algorithms, time-related metrics, energy consumption, and more...
4) There are some minor syntax, grammar and vocabulary errors throughout the manuscript. For example, in the Abstract, the second sentence does not have a subject before the verb.
Reviewer 3 Report
Congratulations for this very interesting paper. To topic is of interest and the presentation is very clear (even though sometimes it may give the impression of too many details and some explanations may seem redundant).
Please find below few remarks/ questions:
In Fig 2c) it might be a notation error: The dotted blue square should be Q2 AD3, not Q1 AD3.
The details relations from pages 9-12 are redundant on each scenario: the second equation per scenario is already written in the first one.
In Table 5 – one question: for all n, when r=8 and r=16, CA7 provides the biggest values, while for r=32, CA6 does it. From Fig 7 (for r=84) is seems CA7 should provide always bigger values. Can you please explain.
Round 2
Reviewer 2 Report
The authors have addressed all of my comments.